# Plant LHC-like proteins show robust folding and static non-photochemical quenching

Petra Skotnicová [1], Hristina Staleva-Musto [2], Valentyna Kuznetsova [2], David Bína [2,3], Minna M. Konert [1], Shan Lu [4], Tomáš Polívka [2,3,5 ✉] & Roman Sobotka [1,5 ✉]

Life on Earth depends on photosynthesis, the conversion of light energy into chemical energy. Plants collect photons by light harvesting complexes (LHC)—abundant membrane proteins containing chlorophyll and xanthophyll molecules. LHC-like proteins are similar in their amino acid sequence to true LHC antennae, however, they rather serve a photoprotective function. How pigments associated with LHC-like proteins are organised and how they contribute to protein function has not yet been determined. Here, we characterize plant LHC-like proteins (LIL3 and ELIP2) produced in the cyanobacterium *Synechocystis* sp. PCC 6803 (hereafter *Synechocystis*). Both proteins were associated with chlorophyll *a* (Chl) and zeaxanthin and LIL3 was shown to be capable of quenching Chl fluorescence via direct energy transfer from the Chl $Q_y$ state to zeaxanthin $S_1$ state. Interestingly, the ability of the ELIP2 protein to quench can be acquired by modifying its N-terminal sequence. By employing *Synechocystis* carotenoid mutants and site-directed mutagenesis we demonstrate that, although LIL3 does not need pigments for folding, pigments stabilize the LIL3 dimer.

[1] Institute of Microbiology, Academy of Sciences of the Czech Republic, Třeboň, Czech Republic. [2] Faculty of Science, University of South Bohemia, České Budějovice, Czech Republic. [3] Biology Centre, Institute of Plant Molecular Biology, Academy of Sciences of the Czech Republic, České Budějovice, Czech Republic. [4] School of Life Sciences, Nanjing University, Nanjing, China. [5] These authors jointly supervised this work: Tomáš Polívka, Roman Sobotka. ✉email: tpolivka@jcu.cz; sobotka@alga.cz

Chlorophyll-binding proteins are the fundamental building blocks of the photosynthetic apparatus in oxygenic phototrophs. As the most abundant cofactor in photosystems, Chl molecules participate in light harvesting and exciton transfer as well as in redox photochemistry. Carotenoid molecules are almost always associated with Chl-binding proteins, primarily to quench the Chl triplet state and, therefore, prevent the formation of reactive oxygen species but also to broaden the spectrum of captured photons[1]. In cyanobacteria, only a handful of proteins are known to bind Chl as a cofactor and these proteins are almost exclusively subunits of photosystems. From a structural point of view, modified Photosystem II (PSII) subunits such as IsiA protein involved in stress responses belong to the same category[2]. However, as an exception to this rule, there is a family of small (one-helix) Chl-binding membrane proteins that show no structural similarity to photosystem subunits. These are the so-called High-light-inducible proteins (Hlips), which are present in virtually all cyanobacteria[3] and contain a characteristic Chl-binding motif (ExxNxR) that probably also serves as a dimerization motif (zipper) buried in the membrane[4]. A dimer of Hlips is capable of binding 4–6 Chls and two carotenoids[4,5] and it is widely accepted that two helices with the ExxNxR motif have to dimerize to create a stable Chl-binding site[4,6]. Notably, pigments associate around the Hlip transmembrane helices in a unique configuration allowing fast quenching (in a few ps) of singlet excited Chls via energy transfer from the Chl $Q_y$ band to the 'forbidden' carotenoid $S_1$ state[5,7]. Although other Chl-binding protein families contain carotenoids in close vicinity to Chl molecules, this type of non-photochemical quenching (NPQ) has been conclusively proven only in Hlips. However, the molecular mechanism that 'opens' the carotenoid $S_1$ state for the fast energy transfer from Chl remains elusive.

In cyanobacteria, Hlips play a vital role in the biogenesis and/or repair of PSII during stress conditions. Hlips seem to interact specifically with the assembly intermediates of PSII, and although these proteins accumulate during stress, they bind only a tiny fraction of cellular Chl[8]. At the onset of the evolution of algae, a small Hlip family radiated into a large and diverse protein superfamily with one to four transmembrane helices[9]. The most abundant descendants of Hlips are light-harvesting complexes (LHCs), which collect photons for photosynthesis in plants as well as in most algae. The static quenching mechanism that evolved originally in Hlips became tuneable in LHC proteins and particularly in those delivering energy to PSII. The switch from harvesting to the dissipative mode in LHCII antennas according to the actual saturation of the photosynthetic machinery is the primary photoprotective mechanism of eukaryotic phototrophs[10].

Apart from 'true' LHCs, a broad spectrum of so-called LHC-like proteins have been identified in algae and plants. LHC-like proteins are, however, much less abundant in plastids than LHC antennas and apparently not involved in light harvesting[9]. ONE-HELIX PROTEINS (OHPs) are most similar to Hlips, both structurally and functionally, as they are also implicated in the biogenesis of PSII[11]. Another type of LHC-like proteins in plants are LIGHT-HARVESTING-LIKE 3 (LIL3) proteins (Fig. 1a). These possess two membrane helices and are known to be essential for the synthesis of phytol[12,13]. Nevertheless, the exact role of LIL3s remains enigmatic and similarly unclear is the role of three-helix EARLY-LIGHT-INDUCED PROTEINS (ELIPs; Fig. 1a). It is believed that ELIPs act as photoprotectants, reducing the damaging effects of high light[14]; however, regulatory and functional roles in Chl biosynthesis have been also reported[15,16].

Although LIL3 and ELIP proteins probably bind Chl molecules in vivo[17,18] and the binding of Chl to LIL3 in vitro has been reported[19,20], how the associated pigments are organized and what exactly is the mechanism of binding are long-standing

questions. In this work, LIL3.1 and ELIP2 proteins from Arabidopsis were heterologously expressed and isolated from the cyanobacterium Synechocystis. Both proteins bind Chl and zeaxanthin (Zea), but only LIL3 exhibited NPQ via Chl $Q_y$ to carotenoid $S_1$ transfer. Nonetheless, the fast energy dissipation can be (re)established in ELIP2 protein by modifying its N-terminus to strengthen carotenoid binding. We further demonstrate that LIL3 can fold in the absence of xanthophylls; however, the pigmentless LIL3 dimer is significantly less stable. We propose a mechanism for the pigment binding of LIL3 and discuss the regulatory function of this protein as a Chl sensor.

## Results

**LIL3 and ELIP2 proteins bind Chl and Zea in vivo**. Biochemical characterization and transient-absorption spectroscopy of LHC-like proteins require highly pure and concentrated samples, which would be challenging to isolate from native sources (plants or algae). We, therefore, decided to heterologously express LIL3.1 (LIL3, 25.1 KDa) and ELIP2 (15.6 KDa) from Arabidopsis in Synechocystis. Genes coding for these two proteins were truncated to remove the signal sequences and an 8×His-tag was added (Supplementary Fig. 1). The genes were then cloned into the Synechocystis genome under a constitutive psbAII promoter (Supplementary Table. 1). The resulting Synechocystis expression strains were cultivated in the presence of ferrochelatase inhibitor to increase the concentration of free Chl in membranes[3], and LIL3 and ELIP2 proteins were purified from the solubilized membranes using a nickel column. The resulting eluates had an intensive green-yellow colour, whereas the control elution from wild type cells was colourless. The isolated LIL3 and ELIP2 proteins were further separated on a sucrose gradient and the fractions containing His-tagged protein were collected (Supplementary Fig. 2).

Analysis of the isolated LIL3 by 2D clear native/SDS electrophoresis (CN/SDS-PAGE) revealed its tight association with pigments (Fig. 1b; Supplementary Fig. 3a). As the LIL3 polypeptide contains only a single ExxNxR motif (Fig. 1a), we predicted that LIL3 binds pigments as a dimer. Indeed, LIL3 showed a tendency to aggregate, however, in order to estimate the mobility of a putative dimer, the mass of dodecyl-β-maltoside (DDM) belt needs to be considered. As no structure is available for LHC-like proteins with two membrane-spanning helices, we predicted the DDM belt mass for ELIP2, the structure of which can be modelled using LHCII as a template. The obtained value for the DDM belt around ELIP2 (~40 KDa; Supplementary Fig. 4) indicates that for a dimeric LIL3 with four membrane-spanning helices, the mass of stably attached detergent is hardly <50 KDa. The majority of LIL3 migrated on CN-gel with a mass between 120–150 KDa, which corresponds to a dimeric LIL3-pigment complex (57 KDa, see later) consisting of a ~75 KDa detergent belt. In contrast to the pigmented dimer, the monomeric form of LIL3 was colourless (Fig. 1b).

The Chl fluorescence of gel-embedded LIL3 protein was very weak, which is indicative of very efficient quenching (Fig. 1b). To confirm this observation, and also to obtain more details about the associated pigments, the purified LIL3 was further analysed by size-exclusion chromatography (SEC) on an HPLC machine equipped with fluorescence and diode-array detectors. The SEC fraction containing LIL3 showed strongly quenched Chl fluorescence when compared with the fluorescence of Chl migrating with DDM micelles, as evident from the ratio of Chl absorbance and Chl fluorescence for LIL3 and DDM peaks (Fig. 1c). This is consistent with the observation of quenching in the CN-gel. Pigments extracted from the LIL3 SEC fraction contained Zea and Chl in a 0.38:1 molar ratio (Fig. 1d). Given the known structure of the Chl-binding motif in LHC proteins with two

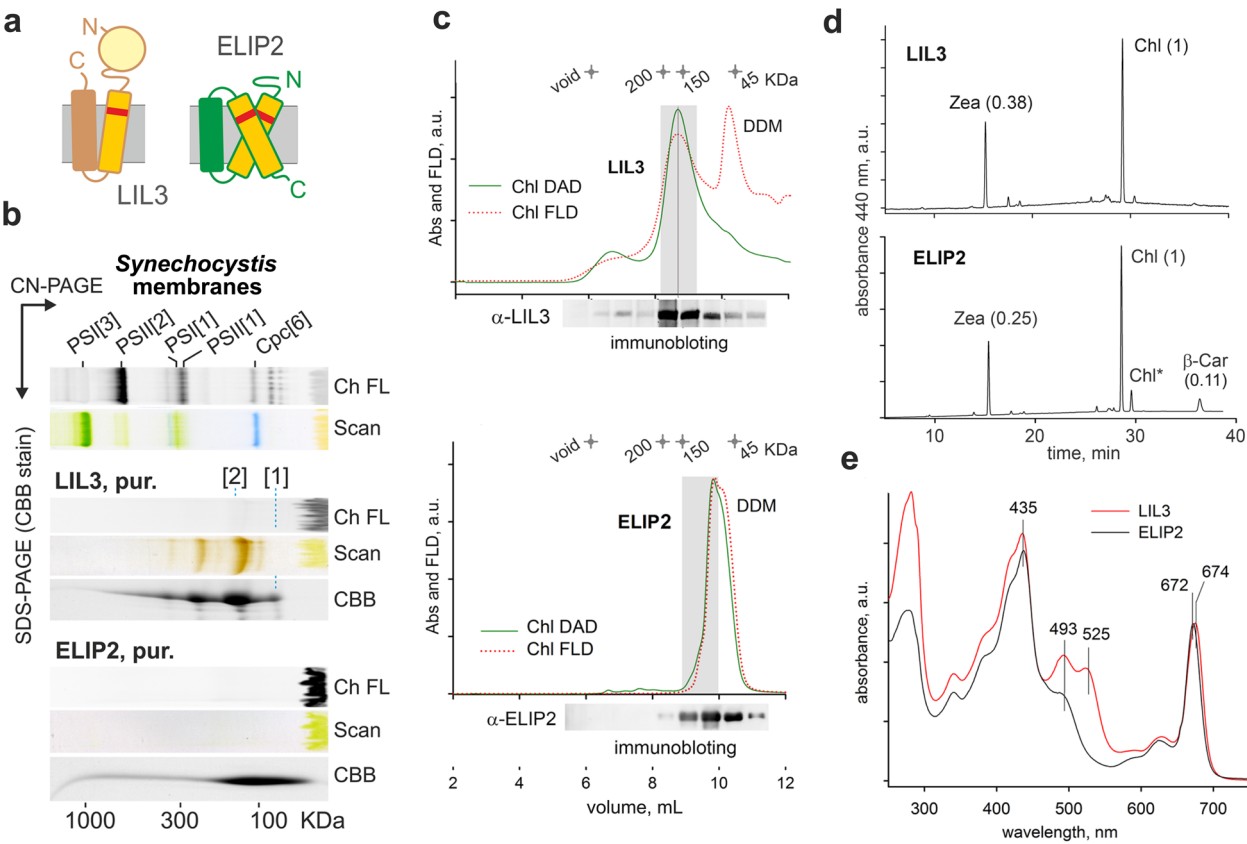

**Fig. 1 Isolation and analysis of LIL3 and ELIP2 proteins. a** A scheme of LIL3 and ELIP2 proteins; the Chl-binding 'Hlip' helix is drawn as an orange rectangle and the ExxNxR motif is highlighted as a red strip. **b** 2D CN/SDS-PAGE of the purified (pur.) LIL3 and ELIP2 proteins after sucrose gradient (Supplementary Fig. 2); 1.4 μg of Chl was loaded for each sample. The native gel was scanned (Scan) and Chl fluorescence (Ch FL) was detected after excitation with blue light. Separated *Synechocystis* membrane complexes (5 μg of Chl) are shown as a 'mass' control: PSI[3]—trimeric PSI (1 MDa), PSI[1] monomeric PSI (300 KDa), Cpc[6]—a hexamer of phycobilisome rod subunits, (100 KDa). LIL3[1] and LIL3[2] indicate monomeric and dimeric LIL3 (see Supplementary Fig. 3 for full-sized gels); PSII[1] and PSII[2]—monomeric and dimeric Photosystem II. Proteins separated in the second dimension were stained with Coomassie Blue (CBB). **c** SEC analysis of purified LIL3 and ELIP2. Chl absorbance and fluorescence of the separated proteins were monitored by diode-array and fluorescence detectors (Chl DAD and Chl FLD, respectively); both channels were normalized to their maxima. Fractions of 0.5 mL were collected, 20 μL of each separated by SDS-PAGE, blotted and probed by specified antibodies. Fractions collected within the grey rectangle were concentrated and used for pigment analysis (see later). **d** Pigments extracted from SEC fractions, indicated in grey, were quantified using HPLC; molar stoichiometries of the identified pigments are shown in parentheses. Values represent the means of three technical replicates; standard deviations were below 10%. Chl* indicates an unknown derivate of Chl. **e** Absorption spectra of LIL3 and ELIP2 proteins as recorded by a DAD during SEC and normalized to the Qᵧ peak.

xanthophyll molecules, we suggest that the dimeric LIL3 binds 2 Zea and 5–6 Chl molecules in an energy-dissipative configuration.

In contrast to the well-resolved LIL3 with pigments on CN-PAGE, the isolated ELIP2 pigment-protein holocomplex dissociated during electrophoresis (Fig. 1b). This result signalled much lower stability of the pigmented ELIP2 than observed for LIL3. Similarly, pigments appear to partially dissociate from the ELIP2 during the SEC. As the protein is eluted together with DDM micelles, the pigmented ELIP2 cannot be resolved well. Nonetheless, the comparison of Chl absorbance and Chl fluorescence of ELIP2 and free Chl in DDM micelles indicated that Chl pigments, co-migrating with ELIP2 during SEC, are not quenched (Fig. 1c). Pigments co-isolated with ELIP2 consisted of Zea, Chl, and β-Car in ratio 0.25:1:0.11, which is very close to carotenoids to Chl molar ratio found for LIL3 (Fig. 1d).

The UV–VIS spectra of LIL3 and ELIP2 proteins were measured during SEC by a diode-array detector (Fig. 1e). It is notable that the spectra of these proteins differ dramatically in the carotenoid region (470–520 nm), despite the similar ratio of associated carotenoid and Chl molecules with very red shifted and intensive absorption of Zea in LIL3 (Fig. 1e). This implies that in LIL3 at least one Zea molecule is tightly bound with a

terminal ring twisted to s-trans configuration[21] while the binding of carotenoids to ELIP2 does not induce the s-trans configuration. It is worth noting that twisting (distortion) of carotene molecule is proposed as a prerequisite for fast NPQ[21,22].

**Replacement of N-terminus strengthens the carotenoid binding to ELIP2.** The absence of any carotenoid twist (red shift) and the detachment of pigments during CN-PAGE (Fig. 1b) indicates weak binding of carotenoids to ELIP2 isolated from *Synechocystis*. In contrast, the binding of carotenoids to LIL3 is tight and in a configuration allowing energy dissipation. Therefore, we decided to modify carotenoid binding in ELIP2 to enhance the carotenoid stability in this protein. Stromal loops play an important role in the coordination of xanthophylls in LHC antennae[23], and thus we replaced the N-terminus in ELIP2 with the N-terminal (stromal) part from LIL3 (Fig. 2a; Supplementary Fig. 1). The resulting chimeric LIL3-ELIP2 protein (Li-ELIP) was expressed in *Synechocystis* and co-purified with pigments (Fig. 2b; Supplementary Fig. 3c). Remarkably, monomeric Li-ELIP co-migrated with pigments on CN-PAGE and strongly quenched Chl fluorescence (Fig. 2b). Due to the significantly larger mass of Li-ELIP (26.6 KDa) compared to ELIP2 (15.6 KDa), the pigmented

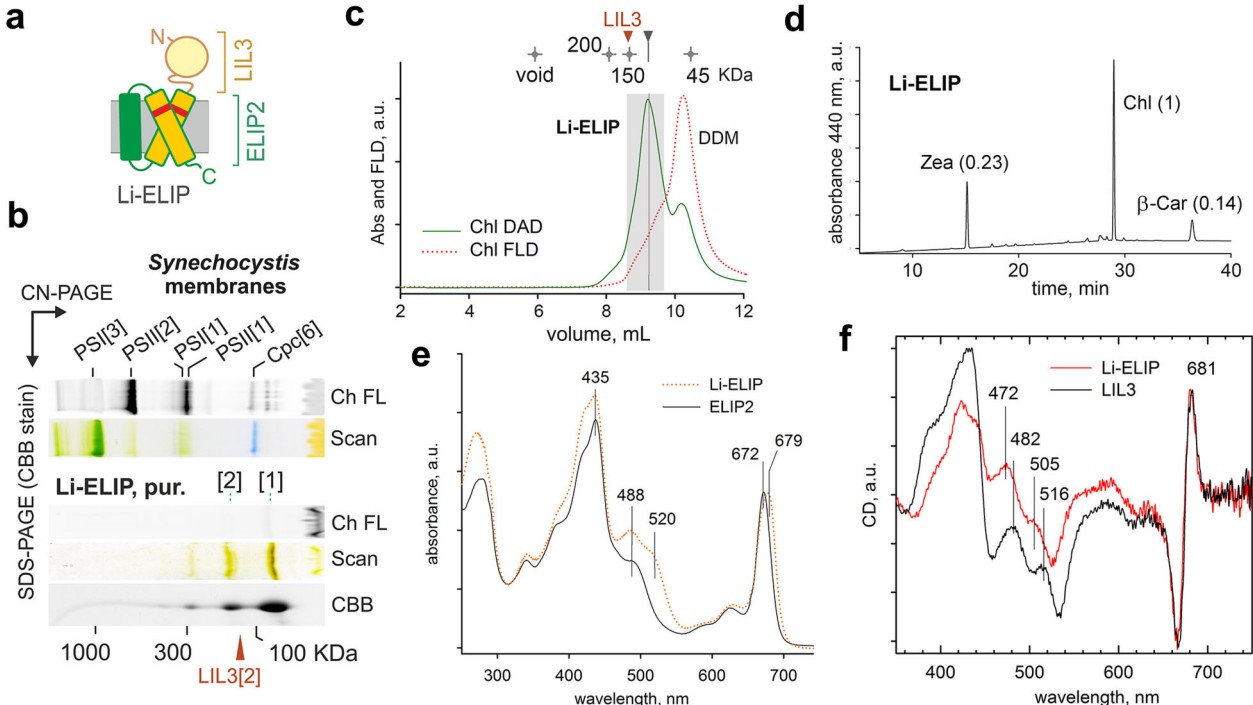

**Fig. 2 Characterization of the isolated Li-ELIP protein. a** A scheme of chimeric Li-ELIP protein; see also Supplementary Fig. 1 for the protein amino acid sequence. **b** *Synechocystis* membranes and the purified Li-ELIP were separated on CN-PAGE, 5 and 1.4 μg of Chl were loaded, respectively. The gel was scanned (Scan) and Chl fluorescence (Ch FL) was detected after excitation with blue light. Proteins were further separated by SDS-PAGE in the second dimension and the 2D gel was stained with Coomassie Blue (CBB). **c** SEC separation of Li-ELIP; Chl absorbance and fluorescence were monitored during chromatography by diode-array and fluorescence detectors (Chl DAD and Chl FLD). Fractions of 0.5 mL were collected, and those shown within the grey rectangle were concentrated and used for pigment analysis. **d** Extracted pigments were analysed by HPLC; molar stoichiometries of the identified pigments are shown in parentheses. Values represent means of three technical replicates; standard deviations were below 10%. **e** Absorption spectra of Li-ELIP and ELIP2 proteins as recorded by a DAD during SEC. **f** CD spectra in the visible region of LIL3 and Li-ELIP proteins. Data were scaled to the CD of the $Q_y$ region at 660–690 nm.

Li-ELIP can be well resolved from free pigments using SEC (Fig. 2c). This analysis confirmed an efficient NPQ in monomeric Li-ELIP. Although the pigment content in Li-ELIP was identical to ELIP2 (Fig. 2d), the carotenoid absorbance and also Chl absorbance were significantly red shifted (Fig. 2e).

To compare the pigment geometry of Li-ELIP and LIL3 proteins, we measured circular dichroism (CD) spectra in the visible region (Fig. 2f). In the spectral region above 630 nm, corresponding to the Chl $Q_y$ band, the CD spectra from each complex are very similar, despite the absorption maximum of the Li-ELIP being red shifted by several nm compared to LIL3 (Supplementary Fig. 5). The CD spectrum consists of a negative peak at 667 nm and positive peak at 681 nm and a zero-crossing point at 675 nm, corresponding to the absorption maximum of Chl in the LIL3 protein. The identity of the CD signal in the Chl spectral region for both proteins strongly suggests a similar geometry of the Chls in LIL3 and those at the blue edge of the Li-ELIP $Q_y$ band. This result also indicates that the pigments at the red shoulder of the Li-ELIP absorption band are not excitonically coupled, hence the origin of the Chl red shift is likely due to pigment-protein interactions.

The similarity of the CD spectra of LIL3 and Li-ELIP extends to the region of the carotenoid absorption as well, the ~8 nm red shift of the LIL3 absorption notwithstanding, pointing to the similarity of the geometry of carotenoids. The most conspicuous feature of the carotenoid CD spectrum is the negative maximum peak at about 10 nm to the red from the lowest energy band in the carotenoid absorption spectrum (Fig. 2f; Supplementary Fig. 5). A less pronounced negative peak (a shoulder in Li-ELIP) is visible about

30 nm to the blue from this main feature. This CD band is more resolved in LIL3, in agreement with the higher resolution of the vibronic bands in the absorption spectrum.

**Energy transfer in LIL3, ELIP2, and Li-ELIP proteins**. We have applied femtosecond transient-absorption spectroscopy to provide details about the origin and dynamics of Chl quenching in LIL3, ELIP2, and Li-ELIP proteins. Given the potential low stability of the purified ELIP2 (Fig. 1b), we omitted the sucrose gradient purification step for this protein; the ELIP2 was thus analysed shortly after its elution from the nickel column. All proteins were excited close to the maximum of the Chl $Q_y$, band and transient-absorption spectra were monitored over the 480–750 nm spectral region. The key data are shown in Fig. 3; full datasets are presented in Supplementary Fig. 6.

Immediately after excitation, the transient-absorption spectrum of LIL3 (Fig. 3a) exhibits characteristic features of an excited Chl: a negative signal peaking around 675 nm representing ground state bleaching, stimulated emission from the excited $Q_y$ band, and weak featureless excited-state absorption in the 480–650 nm region. At 4 ps after excitation, however, the transient spectrum transforms to a shape featuring carotenoid bands associated with the $S_1$-$S_n$ transition (587 nm) and ground state bleaching (528 nm). This clearly shows that energy absorbed by Chl has been transferred to the carotenoid $S_1$ state as it was earlier reported for Hlip[5]. The same scenario occurs for Li-ELIP (Fig. 3b), except the carotenoid ground state bleaching is now at 518 nm and reflects the shift of the carotenoid bands in the absorption spectrum (Supplementary Fig. 5). The kinetics of

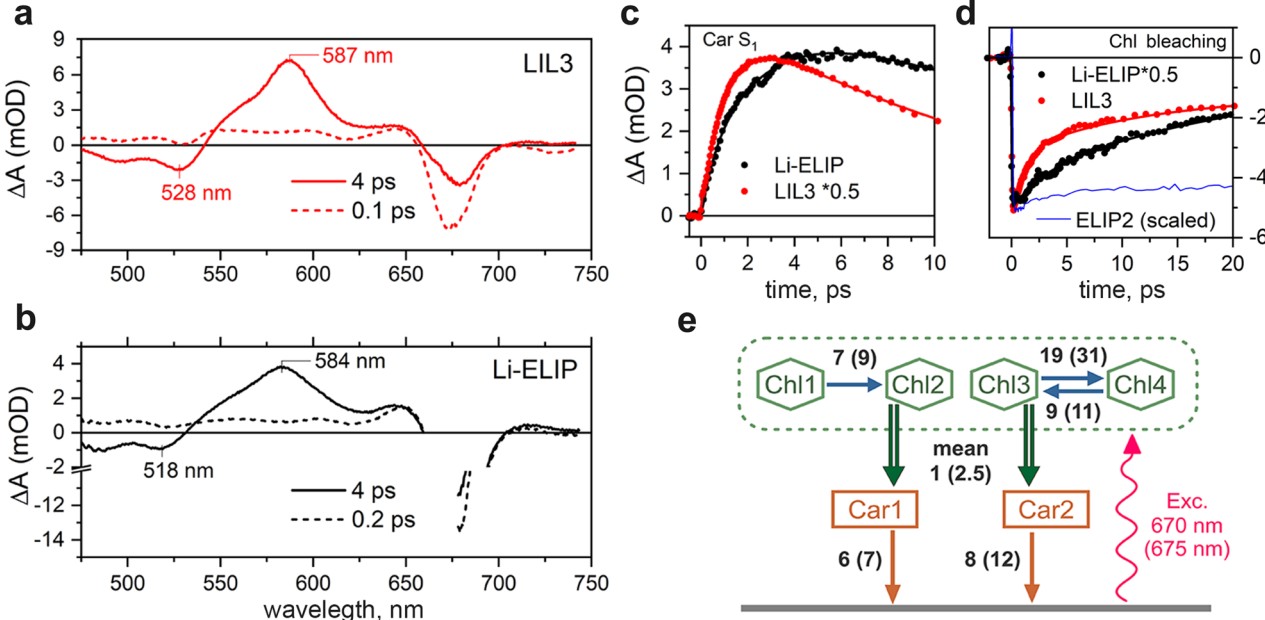

**Fig. 3 Ultrafast transient-absorption data measured after excitation of Chl in LIL3 and Li-ELIP.** Transient-absorption spectra of LIL3 (**a**) and Li-ELIP (**b**) immediately after excitation (dashed) and at 4 ps (solid). LIL3 and Li-ELIP proteins were excited at 670 nm and 675 nm, respectively. **c** The rise of the carotenoid signal after Chl excitation was measured at the maximum of the $S_1$–$S_n$ band at 587 nm for LiL3 and 584 nm for Li-ELIP. **d** Decay of excited Chl monitored at 685 nm; the blue line shows Chl dynamics for unquenched ELIP2 for comparison. **e** Unified scheme of energy transfer pathways in LIL3 and Li-ELIP. The model assumes non-selective excitation of a pool of four Chl (red wavy arrow). The green double arrows represent the main Chl-to-carotenoid energy transfer quenching channel. Solid arrows denote relaxation processes associated either with the equilibration between Chl pools (blue) or with the decay of the carotenoid $S_1$ state (orange). The numbers correspond to the time constants (in ps) associated with each process in LIL3 (Li-ELIP). See Supplementary Fig. 8 for details.

**Table 1 Time constants and relative contributions at 680 nm of Chl decay components obtained in LIL3 and Li-ELIP following excitation into $Q_y$ band of Chl.**

| LIL3 | $\tau$ [ps] | 1.1 | 11 | 39 | >5000 |
|---|---|---|---|---|---|
| | A (rel.) | 0.44 | 0.26 | 0.07 | 0.23 |
| Li-ELIP | $\tau$ [ps] | 2.4 | 17 | 56 | >5000 |
| | A (rel.) | 0.21 | 0.52 | 0.2 | 0.07 |

energy transfer quenching are given in Fig. 3c and show the appearance of the signal associated with the carotenoid $S_1$ state after Chl excitation. For ELIP2, no fast decay of Chl signal has been observed (Fig. 3d), confirming the assumption that Chl in ELIP2 is not quenched. This conclusion is further corroborated by the full dataset shown in Supplementary Fig. 6, which proves the absence of any carotenoid signal after Chl excitation of ELIP2.

To explore the Chl quenching dynamics in LIL3 and Li-ELIP in detail, we have applied global fitting analysis. The application of a sequential fitting model to both proteins revealed multiexponential dynamics of the Chl excited state that could be satisfactorily described with four components (Supplementary Figs. 7, 8). The slowest has a lifetime of >5 ns and apparently corresponds to free Chl. This component is more prominent in LIL3, comprising about 20% of the Chl decay at 680 nm, compared to about 3–7% in Li-ELIP (Table 1). The three faster components are associated with quenching; their time constants (and relative contributions at 680 nm) were ~1 ps (0.44), 11 ps (0.26) and 39 ps (0.07) for LIL3, ~2.5 ps (0.21), 17 ps (0.52) and 56 ps (0.20) for Li-ELIP (Table 1). Thus, while the fastest energy transfer channel dominates in LIL3, the Chl decay in Li-ELIP is determined by the intermediate ~10–20 ps channel as well as an increased proportion of the slowest channel. Omitting the >5 ns component

that is likely due to contamination by free Chl, the average lifetime of Chl excited state due to quenching is ~8 ps in LIL3 and ~20 ps in Li-ELIP. The slower decay of excited Chl in Li-ELIP compared to LIL3 can be clearly appreciated in Fig. 3d. It should be noted that the average lifetime of Chl increased towards the red edge of the Chl bleaching, in particular in Li-ELIP, where it was ~25 ps at 685 nm, while it remained at ~8 ps at this wavelength in LIL3. Hence, it appears that one factor contributing to the slower quenching in Li-ELIP is the larger contribution of the red-Chl forms that slow down the energy equilibration within the Chl pool. Nevertheless, the observed lifetimes in both LIL3 and Li-ELIP correspond to quenching efficiency close to 100%, assuming free Chl lifetime of 5 ns.

Based on these results we have constructed a model of the energy transfer network applicable to both LIL3 and Li-ELIP (Fig. 3e). The model is able to reproduce well the measured data and the reconstructed decay associated difference spectra/ evolution associated difference spectra (DADS/EADS) match those obtained from direct sequential fitting (Supplementary Fig. 8). The model features four compartments associated with Chl and two carotenoids, however, only two Chl are involved directly in the energy transfer quenching channel between the Chl $Q_y$ states. Their dynamics are characterized by average time constants of 1.1 and 2.4 ps in LIL3 and Li-ELIP, respectively. The slower channels are associated with equilibration between Chl pools and they affect the overall dynamics, especially in Li-ELIP.

**The affinity of LIL3 and Li-ELIP proteins for violaxanthin.** Although the thylakoid membranes of cyanobacteria and plants are similar regarding the lipid content, plant LHC antennas bind xanthophylls such as lutein and violaxanthin (Vio) that are not present in cyanobacteria[24]. Whereas lutein is chemically close to Zea and LHCII proteins show similar affinities for these two

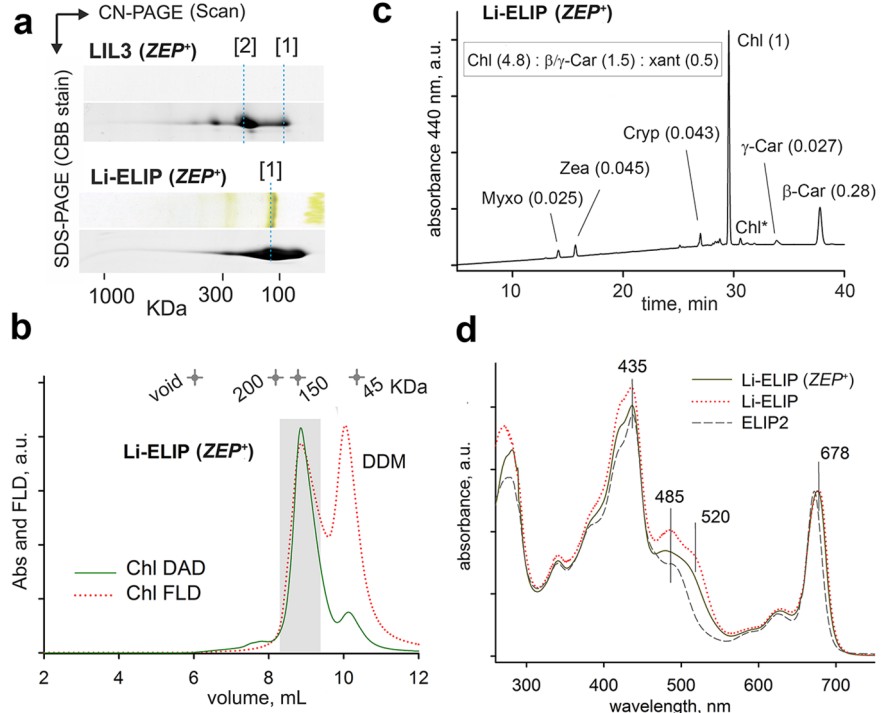

**Fig. 4 Analysis of the LIL3 and Li-ELIP proteins co-expressed with ZEP enzyme (_ZEP_+). a** The purified LIL3 (_ZEP_+) and Li-ELIP (_ZEP_+) protein variants were separated on 2D CN/SDS-PAGE; [1] and [2] indicate monomeric and dimeric forms, respectively. **b** SEC separation of Li-ELIP; Chl absorbance and fluorescence were monitored during chromatography by diode-array and fluorescence detectors (Chl DAD and Chl FLD). Fractions of 0.5 mL were collected and those shown within the grey rectangle were concentrated and used for pigment analysis. **c** Extracted pigments were analysed by HPLC; molar stoichiometries of the identified pigments are shown in parentheses. Values represent the means of three technical replicates; standard deviations were below 10%. A probable variant of pigment content per monomeric Li-ELIP protein is shown as an inset. **d** Absorption spectra of Li-ELIP (_ZEP_+), Li-ELIP, and ELIP2 proteins.

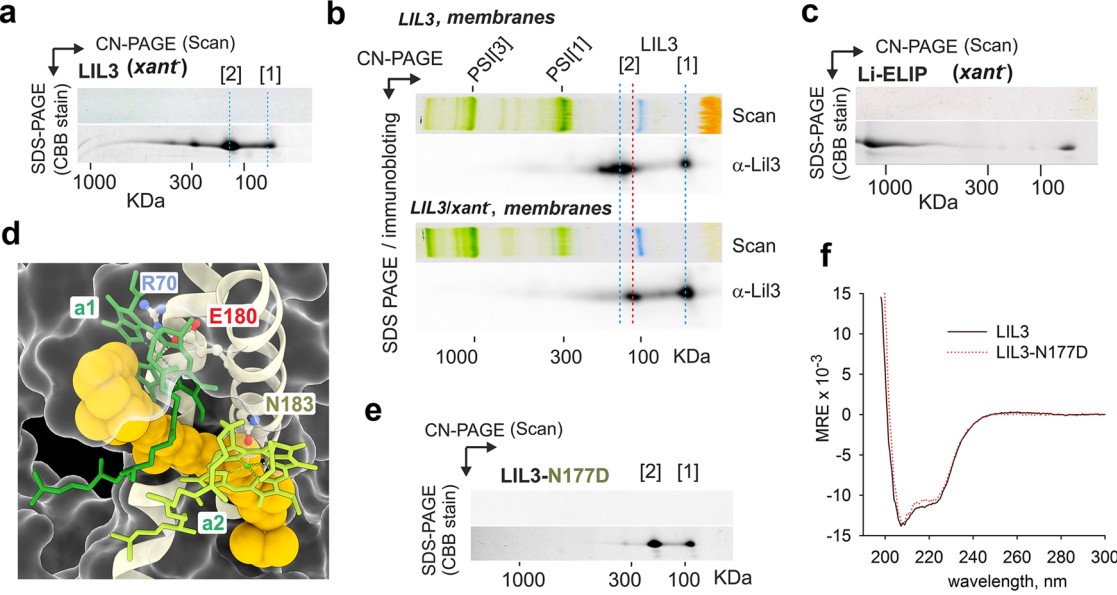

**Fig. 5 Carotenoid specificity, dimerization, and mutagenesis of LHC-like proteins produced in _Synechocystis_. a** LIL3 was expressed in the _Synechocystis_ _xant_− background containing synechoxanthin and β-Car as the only carotenoids (see Supplementary Fig. 9a). Purified LIL3 was separated by 2D CN/SDS-PAGE and the gel stained by Coomassie Blue (CBB). **b** Solubilized membrane proteins isolated from the _Synechocystis_ strain expressing LIL3 (_LIL3_ strain) and from the _LIL3/xant_− were separated by 2D CN/SDS-PAGE. The SDS-gel was blotted, and the LIL3 protein was detected by a specific antibody. **c** Li-ELIP purified from the _xant_− background was separated by 2D CN/SDS-PAGE and the gel stained by CBB. **d** A detail of lutein (_lut_1) binding site in pea LHCII protein (PDB code 2BHW[23]). Chl molecules coordinated by E180 (Chl _a_1, forest green) and N183 residues (Chl _a_2, light green) close the lutein-binding cavity. Lutein is depicted as a yellow, space-filling model. **e** The purified LIL3-N177D protein was separated by 2D CN/SDS-PAGE, showing that the N177D mutation abolishes pigment binding. N177 residue in LIL3 corresponds to N183 Chl ligand in LHCII. **f** CD spectra in the UV region of LIL3 and the N177D LIL3 mutant normalized per residue (MRE mean residue ellipticity).

xanthophylls (see Discussion), the epoxy-carotenoid Vio is chemically distant to the carotenoids in cyanobacteria. It is indeed an important question whether LIL3 or ELIP2/Li-ELIP can bind Vio, either exclusively or together with Zea. Therefore, we co-expressed LIL3 and Li-ELIP proteins in *Synechocystis* together with the plant Zea epoxidase enzyme (ZEP) that converts Zea into Vio. A small amount of antheraxanthin, a ZEP intermediate, can be also detected in the $ZEP^+$ strain (Supplementary Fig. 9a).

The LIL3 protein isolated from the $ZEP^+$ background appeared colourless; however, it still migrated in the CN-gel mostly as a dimer (Fig. 4a; Supplementary Fig. 10a). When the LIL3/$ZEP^+$ eluate was concentrated 20 times, the solution became brownish and the extracted pigments contained Zea and Chl in a 0.55:1 ratio (Supplementary Fig. 9b). This result implies that LIL3 does not bind Vio, rather has a high affinity to Zea and a small fraction of LIL3 can associate with pigments even though the content of Zea in membranes is very low (Supplementary Fig. 9a). In contrast to LIL3, the Li-ELIP/$ZEP^+$ protein co-eluted from the nickel column with a visible quantity of pigments (Fig. 4a) and quenched Chl fluorescence, which we confirmed by SEC analysis (Fig. 4b). The spectrum of pigments associated with Li-ELIP/$ZEP^+$ was very unusual; apart from Zea and β-Car, we found myxoxanthophyll, cryptoxanthin, and γ-Car, but no Vio or antheraxanthin (Fig. 4c). The flexibility in carotenoid binding to Li-ELIP is intriguing. This artificial protein can be associated even with myxoxanthophyll containing a sugar moiety and, in contrast to LIL3, it can accommodate β-Car as the main carotenoid without losing the capability to quench. The carotenoid absorbance of Li-ELIP/$ZEP^+$ is less intense around 485 nm when compared with Li-ELIP; nonetheless, it is clearly more red shifted than in ELIP2 (Fig. 4d).

**Folding and oligomerization of LIL3 without associated pigments.** LHC antenna proteins have never been isolated or reconstituted without xanthophylls[25]; however, the LIL3 purified from the $ZEP^+$ background was almost colourless (Fig. 4a). This finding indicated that LIL3 is stable and folded without any pigments bound. To exclude that transient or weak interactions with xanthophylls promote LIL3 folding in vivo, we prepared a $xant^-$ *Synechocystis* strain, which lacks all xanthophylls except for a small amount of synechoxanthin (Supplementary Fig. 9a). LIL3 purified from the $xant^-$ genetic background was colourless and behaved on CN-PAGE as a dimer (Fig. 5a), consistent with the LIL3/$ZEP^+$ variant (Fig. 4a). The oligomerization of LIL3 might be, however, an artefact of purification (high protein concentrations). Therefore, we inspected the presence of LIL3 oligomers in the solubilized membranes using 2D CN/SDS-PAGE combined with immunodetection. In wild type membranes, LIL3 is mostly a dimer, and a dimeric form of LIL3 can be detected also in $xant^-$ membranes, although its level is significantly reduced (Fig. 5b). In the latter case, the dimeric LIL3 shows a shift in mobility, most likely due to missing pigments. It is also worth noting that we detected no other high order oligomers of LIL3 than a dimer in the membranes (Fig. 5b). Higher-order LIL3 aggregates (>200 KDa) observed on CN-PAGE (Fig. 1b) are thus an artefact of sample concentration and/or the CN-PAGE.

We conclude that without xanthophylls, the LIL3 protein appears correctly folded, but its oligomerization is impaired and Chl molecules do not associate stably. Similarly, Li-ELIP can be purified from the $xant^-$ background as a colourless protein although with a strong tendency to aggregate (Fig. 5c). In LHC antenna proteins, each central xanthophyll forms a compact structure with two Chls ligated by the ExxNxR motif. Chl phytol chains enclose the carotenoid-binding cavity (Fig. 5d), and it is very likely that carotenoids also play a critical role in stabilizing the ligated Chl[26]. To further address the mechanism of pigment

binding to LIL3, we replaced the glutamine residue in the LIL3 ExxNxR motif (Supplementary Fig. 1a) with either Ala or Asp, residues that cannot ligate Chl at the same position in LHCII[27]. Both mutated LIL3 variants (LIL3-N177A and LIL3-N177D) were purified from *Synechocystis* as completely colourless proteins (Fig. 5e; Supplementary Fig. 11). The essentially identical UV CD spectra of the LIL3 and the mutated LIL3-N177D shown in Fig. 5f indicate that the LIL3 does not require pigment binding for the formation of secondary structures.

## Discussion

The recent advances in cryo-EM technology have led to the structural determination of LHC antennas of both PSI and PSII types from evolutionary distant eukaryotes such are green and red algae or diatoms[28,29]. Despite different carotenoids and Chl cofactors in these proteins, the structure of two central inter-twined helices with four Chls ligated by the ExxN/HxR motif remains highly conserved. In addition, in all types of LHCs with solved structure, two xanthophyll molecules can be always recognized in cross-braced *lut*1 and *lut*2 binding sites with their isoprenoid chains wrapped up by phytol tails of *a*1 (*lut*1) and *a*4 Chl (*lut*2; see Fig. 5d for the *lut*1 binding site).

We expect that this six-pigment arrangement reflects a primordial configuration that evolved with Hlips and remained preserved throughout the whole LHC superfamily. Qualitatively, the CD spectra of the LIL3 and Li-ELIP proteins (Fig. 2f) in the Chl Soret and carotenoid spectral region strongly resemble the spectrum of monomeric plant LHCII complex[30], although significantly shifted to lower energies. Additional pigment-binding sites in LHCs are less conserved or not conserved at all[31]. Apart from the central helix pair, stromal/lumenal loops and the third helix in LHC(-like) proteins are also very divergent and perhaps of a different origin[9]. Thus, it is very likely that the dimeric LIL3 and the monomeric ELIP2 contain the conserved set of four Chl and two carotenoids. According to the determined Chl: Zea ratio (Fig. 1d), both proteins should bind one more (fifth) Chl that might be coordinated by an associated lipid as is common with LHCs[32].

In their native environment both LIL3 and ELIP2 bind most likely lutein rather than Zea. In a pioneering isolation of a native ELIP from pea, the final ELIP-enriched fraction contained only lutein and Chl but no other carotenoids or Chl-*b*[18]. Indeed, there is no strong binding preference for lutein or Zea in plant LHCII proteins[33], and LHCIIs can accumulate in Arabidopsis mutants synthesizing either Zea or Vio as the only xanthophyll[34,35]. In this sense, LHC-like proteins appear more selective than LHCII as Vio is accepted for neither of their xanthophyll-binding sites (Fig. 4).

The smallest *Synechocystis* Hlip (HliC, 5 KDa) has been described as a dimer binding four Chl and two β-Cars[4] representing thus a minimal stable pigment configuration as suggested earlier. HliC is capable of a fast quenching via a red-shifted carotenoid[4,7]. Intriguingly, only one of two predicted β-Car molecules is twisted in such a way as to allow a fast Chl $Q_y$ to β-Car $S_1$ energy transfer[21]. What mechanism causes this local carotenoid distortion remains enigmatic. In the LIL3 protein, Zea is extremely red shifted and, indeed, the protein is an even faster quencher than Hlips[5] (Fig. 3), which contrasts sharply with no obvious red shift and no NPQ in the purified ELIP2. These observations are in line with the increased absorption at 535 nm in LHCII after switching to a quenched conformation[36] and further support a crucial role of carotenoid distortion for the fast energy dissipation[22]. Although energy can be theoretically dissipated from LHC proteins without the need for a distorted carotenoid molecule[37], these quenching pathways are around two orders of magnitude slower than the observed values for Hlips[5] or LIL3 (Fig. 3e).

According to our data, there are two Zea molecules per LIL3 dimer or for ELIP2 bound comparably to luteins in LHC antennas (Fig. 5d). The central (*lut*1/2) luteins in LHCI/II, as well as Zea molecules in LHCR present in red algae, are stabilized by their tight contact with stromal loops, typically via hydrogen bonds with Asp, Asn, or Pro residues[29]. ELIPs accumulate under various stress conditions and, given their photoprotective role[14], we assume it is very unlikely that ELIP2 binds Chl in an unquenched configuration in vivo. We hypothesize that the coordination of Zea by the stromal site is weak in the isolated ELIP2, either because Zea is not a genuine component and/or the site is destabilized by DDM. In an attempt to strengthen the xanthophyll binding, we modified the ELIP2 protein to contain the N-terminal stromal region of LIL3 (Fig. 2a). The resulting Li-ELIP protein contains red-shifted Zea (Fig. 2e) and quenches comparably to LIL3 (Fig. 3), although the ratio of pigments remained unchanged (Fig. 2d). This observation indicates that the N-terminal part from LIL3 close to the Zea stromal end ring is responsible for its twisting to s-trans configuration and alters the local structure of Li-ELIP in a way allowing fast NPQ.

LHC antennas are unstable in the absence of xanthophylls; at least one xanthophyll-binding site (*lut*1) must be occupied otherwise, LHCs cannot fold[25]. According to the accepted model, xanthophylls are necessary for the formation of Chl-binding pockets in LHCs[38]; Chl can bind only transiently to LHCII in the absence of carotenoids[26]. Xanthophyll binding, therefore, should precede Chl-binding; nonetheless, the attachment of xanthophylls to LHCs in the absence of Chls is apparently also labile[26]. In contrast to LHCs, LIL3 does not need pigments for stability, and secondary structures in LIL3 are folded without pigments (Fig. 5f). However, we propose that the mechanism of pigment binding to LHC-like proteins as well as to Hlips resembles LHCII (Fig. 6). After binding of two carotenoids and four Chl, almost any flexibility of the central helix pair is attenuated[39], and the resulting rigid structure is very stable. Replacement of the N177 residue in LIL3 abolished pigment binding (Fig. 5e), suggesting that two Chl only (*a*1 and *a*4) are not sufficient to stabilize Zea in the dimeric LIL3. In LHCs, the removal of either *a*2 or *a*5 Chl does not affect the overall stability of the protein[40]; however, to

our knowledge, there is no reported LHC mutant protein lacking both these Chls.

The stabilizing effect of pigments (Chls) on the dimeric structure of LIL3 can be functionally related to the essential role of LIL3 in the synthesis of phytol in plants[12,13]. In Arabidopsis, chloroplast LIL3 is mostly a dimer associated with one or two copies of geranylgeranyl reductase (GGR) and, in the absence of LIL3, the level of GGR as well as of phytol is drastically reduced. Accumulation of GGR, but not its activity, can be fully restored by fusing GGR with a randomly selected membrane-spanning helix[41]. This finding implies that LIL3 prevents degradation of GGR by tethering it onto the membrane but also supports the activity of the enzyme, for instance, by increasing the affinity for the substrate or an electron donor. Consistent with our results, LIL3 harbouring a mutated ExxNxR motif accumulates in chloroplasts and the level of GGR remains also high. However, the dimerization of the mutated LIL3 is almost abolished, and the mutant plants are severely depleted in phytol[41]. We speculate that dimeric LIL3 provides a scaffold for GGR and other potential interacting partners[42] and so keeps the synthesis of phytol high. LIL3 is present in barley etioplasts and probably binds the first pool of Chl produced after illumination[17]. It is therefore possible that the Chl-binding to LIL3 activates GGR to synthesize large quantities of phytol for the building of thylakoid membranes during photomorphogenesis. The mechanism of Chl sensing via stabilizing the 'Hlip' dimer has been already proposed for the ferrochelatase enzyme. In oxygenic phototrophs, this enzyme possesses a typical Hlip helix and forms a stable dimer only when associated with Chl and carotenoids[3]. As in LIL3, the bound Chl is efficiently quenched via energy transfer to the $S_1$ state of a carotenoid nearby.

## Methods

**Construction and cultivation of *Synechocystis* strains.** *Synechocystis* sp. PCC 6803 substrain GT-P[43] was used as the wild type and as a genetic background for all prepared strains (listed in Supplementary Table 1). cDNA of Arabidopsis *LIL3.1* (At4g17600) and *ELIP2* genes (At4g14690) was obtained as a gift from Prof Bernhard Grimm (Humboldt University). Genes without the predicted signal sequences (Supplementary Fig. 1) were re-cloned into pPD-*NFLAG* plasmid[44] using *NdeI* and *BglII* restriction sites; 8xHis sequence was inserted at the N-terminus using a forward PCR primer. The resulting pPD-*LIL3* and pPD-*ELIP2* plasmids were transformed into *Synechocystis* wild type and the obtained transformants fully segregated using increasing concentrations of antibiotics up to 50 mg/mL. Chimeric *Li-ELIP* gene was synthesized by Genscript (USA) and cloned into pPD-*NFLAG* with 8xHis-tag essentially as described for the *LIL3* gene.

The Vio-producing strain *ZEP*+ is described in ref. [45], *xant*− (Δ*cruF*/Δ*crtR*/ Δ*crtO*) was prepared by combining available deletion constructs[46,47]. As *ZEP*+ and *xant*− strains contained a kanamycin resistance cassette (the same selection gene as in the pPD-*NFLAG* plasmid) we produced, therefore, also pPD-*LIL3*, pPD-*ELIP2*, and pPD-*Li-ELIP* plasmid variants with chloramphenicol or erythromycin resistance cassettes. These modified constructs were transformed and segregated in the *ZEP*+ and *xant*− genetic backgrounds (Supplementary Table 1). In order to prepare *LIL3*-N177A and *LIL3*-N177D strains, the pPD-*LIL3* plasmid was modified using the QuikChange kit (Agilent) and transformed into wild type. Sequences of primers used in this study are shown in Supplementary Table 2.

*Synechocystis* strains were grown in liquid BG-11 medium at 28 °C in 1 L cylinders bubbled with air. Cells were first cultivated under a moderate irradiance of 50 μmol photons m$^{-2}$ s$^{-1}$ given by white fluorescence tubes. After reaching an optical density ~0.4 at 750 nm, the culture was supplemented with 250 nM N-methyl mesoporphyrin IX for an additional 18 h to stimulate Chl biosynthesis, and the light intensity was increased to 300 μmol photons m$^{-2}$ s$^{-1}$ to increase the expression from the *psbAII* promoter[3].

**Purification of His-tagged proteins.** For the purification of His-tagged LHC-like proteins, 3.5 L of cells were broken using glass beads (0.1 mm diameter) in buffer A containing 25 mM Na–Phosphate buffer, pH 8.0, 50 mM NaCl, 10% glycerol, and EDTA-free protease inhibitor (Sigma). The pelleted membrane fraction, prepared essentially as described in ref. [48], was resuspended in buffer A (~0.5 mg Chl/mL) and solubilized for 60 min at 10 °C with 1% DDM. Finally, insoluble contaminants were removed by centrifugation (47,000 × g, 20 min). The concentration of NaCl in the supernatant was adjusted to 0.5 M, supplemented with 10 mM imidazole, and incubated with 1 mL of Protino Ni–NTA agarose resin (Macherey-Nagel, Germany) for 60 min. Afterwards, the supernatant with resin was loaded onto an empty plastic

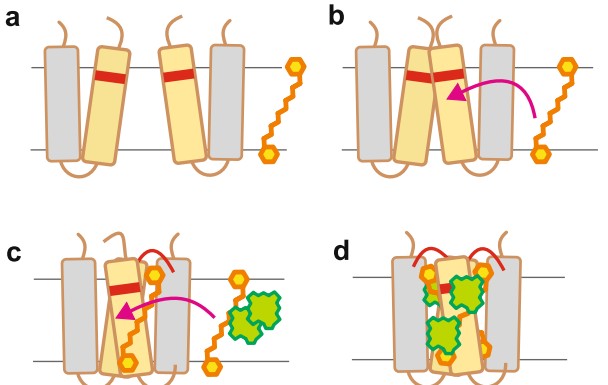

**Fig. 6 A scheme of pigment binding to LIL3 protein.** LHC-like proteins, which contain only a single 'Hlip helix' (Hlips, OHPs, SEP/LIL3) have to dimerize to be able to associate with pigments. Although LIL3 is stable as a monomer (**a**) and can dimerize without pigments, the pigmentless dimer readily dissociates (**b**). A transient attachment of carotenoid molecules to LIL3 dimer is a precondition for the Chl-binding; in the absence of xanthophylls, Chl molecules cannot be ligated by the ExxNxR motif (**c**). On the other hand, the associated Chls enclose the carotenoid-binding cavity (see also Fig. 5d), and therefore the successive binding of carotenoids and Chls on each side of the dimer is essential for the establishment of a stable pigment-protein complex (**d**).

chromatography column (Econo-Pac, Bio-Rad) for washing steps. Proteins bound to the column were washed first with 20 mL of buffer A containing 0.5 M NaCl, 10 mM imidazole, and 0.04% DDM, then 20 mL of buffer A containing 20 mM imidazole and 0.04% DDM and finally with 10 mL of 40 mM imidazole with 0.04% DDM. Proteins were eluted using 6 mL of buffer A containing 200 mM imidazole and 0.04% DDM and concentrated ~12× on Amicon 50 kDa micro-concentrators (Millipore). The eluate was concentrated and 300 µL loaded onto the top of 20% linear sucrose gradients prepared in buffer B (25 mM MES-NaOH pH 6.5, 10 mM MgCl₂, 10 mM CaCl₂) and supplemented with 0.04% DDM. Gradients were prepared by freezing 12 mL of the 20% sucrose solution in Beckman Ultra-Clear centrifuge tubes (14 × 95 mm) at −80 °C for >1 h and thawing at 10 °C. Gradients were centrifuged for 18 h at $285,000 \times g$, + 4 °C in Optima XPN-90 Ultracentrifuge (Beckman Coulter) equipped with SW 40 Ti rotor. After ultracentrifugation, a ~0.7 mL fraction containing the His-tagged proteins was collected and concentrated on Amicon 50 kDa micro-concentrators to ~200 uL.

**2D clear native/SDS electrophoresis and immunoblotting.** For native electrophoresis, the purified protein complexes were separated on 4–14% (w/v) polyacrylamide CN gels in the first dimension according to ref. [49]. The gels were scanned between sheets of plastic foil using an office scanner (Epson Perfection V550), and the Chl fluorescence image was recorded by a LAS-4000 camera (Fuji) after excitation with a blue laser. The individual components of the protein complexes were resolved by incubating the gel strip from the first dimension in 25 mM Tris/HCl pH 7 containing 1% (w/v) SDS and 1% (w/v) dithiothreitol for 30 min at room temperature and by subsequent separation in the second dimension by SDS electrophoresis in a denaturing 12–20% (w/v) polyacrylamide gel containing 7 M urea. The separated proteins were either stained with Coomassie Brilliant Blue or transferred (3 mA/cm2, 3 h) onto an Immobilon-P membrane (0.45 µm; Millipore). After electrotransfer, the membrane was blocked by 0.05% (w/v) Tween 20 in 10 mM Tris/HCl, pH 7.6, 150 mM NaCl for 30 min and subsequently incubated with a specific primary antibody and then with a secondary antibody–horseradish peroxidase conjugate (Sigma, catalogue number A6154; 1: 10,000 dilution). The peroxidase activity was visualized by incubating the membrane for 20 s in Luminata Crescendo Western HRP substrate (Sigma) and detected using LAS-4000 (Fuji). The primary antibodies used in this study were as follows: anti-LIL3.1 antibody (1: 5000) raised in rabbit against the Thr-50 to Lys-84 recombinant fragment of the Arabidopsis protein[50]; anti-ELIP2 antibody (1: 5000) raised in rabbit again the synthetic peptide AQGDPIKEDPSVPSC[51].

**Gel filtration and pigment analysis.** Purified proteins were immediately injected onto an Agilent-1200 HPLC system and separated on a Yarra SEC-3000 column (Phenomenex) using mobile phase (20 mM MES, pH 6.5, 100 mM NaCl, pH 6.5, containing 0.03% (w/v) DDM) at a flow rate of 0.25 ml min⁻¹ at 15 °C. Eluted proteins and complexes were detected using a diode-array detector and a fluorescence detector set to 440/675 nm (excitation/emission wavelengths).

To determine molar stoichiometries of co-eluted pigments, fractions collected from SEC were pooled, and 100 µL injected into Agilent-1260 HPLC system equipped with a diode-array detector. Pigments were separated on a reverse-phase column (Zorbax Eclipse C18, 5 µm particle size, 3.9 × 150 mm; Agilent) with 35% (v/v) methanol and 15% (v/v) acetonitrile in 0.25 M pyridine (solvent A) and 20% (v/v) methanol, 20% (v/v) acetone, 60% (v/v) acetonitrile as solvent B. Pigments were eluted with a linear gradient of solvent B (30–95% (v/v) in 25 min) in solvent A followed by 95% of solvent B in solvent A at a flow rate of 0.8 ml min⁻¹ at 40 °C. Chl and carotenoids were detected at 440 nm, the obtained peaks were integrated, and the molar stoichiometries calculated from calibration curves were prepared using authentic standards.

**Circular dichroism spectroscopy.** CD spectra were collected using the Jasco-715 (Jasco, Japan) instrument. Samples were placed in quartz cuvettes with a path length of 3 mm (pigment CD in the visible range) or 1 mm (protein CD in the UV region). Spectrum measurement bandwidth was set to 2 nm.

**Femtosecond spectroscopy.** Ultrafast transient-absorption data were collected using a femtosecond spectrometer based on a Ti:sapphire regenerative amplifier (Spitfire Ace-100F, Spectra-Physics, USA) seeded with a Ti:sapphire oscillator (MaiTai SP, Spectra-Physics, USA), and pumped by Nd:YLF laser (Empower 30, Spectra-Physics, USA). The laser system produces ~100 fs pulses centred at 800 nm with a 1-kHz repetition rate. The excitation pulses were generated by an optical parametric amplifier (TOPAS Prime, Light Conversion, Lithuania). The probe pulses were generated by focusing a fraction of the 800 nm beam to a 2 mm sapphire plate to generate a broadband (450–750 nm) white light, which was split by a broadband 50/50 beam splitter to reference and probe beams. The probe beam was focused by a 300 mm spherical mirror to the sample where it overlaps with the excitation beam. The probe and reference beams were then focused to the entrance slit of a spectrograph, where they were dispersed onto a double CCD array allowing measurements of transient spectra in a spectral window of ~250 nm. The time delay between the excitation and probe pulses was introduced by a computer-controlled delay line. The mutual polarization of the excitation and probe beams were set to the magic angle (54.7°) by placing a polarization rotator in the excitation beam. For all measurements, a 2 mm path length quartz cuvette was used. To avoid sample degradation, we employed a micro stirrer that continuously mixed the sample during the measurements. Global analysis of the transient-absorption data was performed using Glotaran[52], MATLAB package was used for modelling the excited-state dynamics.

**Statistics and reproducibility.** LIL3, ELIP2, and Li-ELIP proteins were purified at least three times each; we did not observe any significant variability in pigment content among individual preparations. 2D-gel separations, SEC, and the analysis of SEC fractions were performed once for the each protein variant. 2D gel combined with immunodetection of LIL3 in isolated membranes (Fig. 5b) was performed once. The ultrafast spectroscopy data point for each time delay was generated by averaging of 300 laser shots, the whole time profile consisted of averaging 20 time scans, resulting in 6000 laser shots averaged for each data point.

**Reporting summary.** Further information on research design is available in the Nature Research Reporting Summary linked to this article.

## Data availability

The data that support the findings of this study are available from the corresponding authors upon reasonable request. Source data are provided with this paper.

## Code availability

MATLAB scripts, used for the modelling of excited-state dynamics, are available upon reasonable request.

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

## Acknowledgements

P.S., M.M.K. and R.S. are supported by European Research Council Synergy Award 854126. T.P., D.B. and V.K. thank to the Czech Science Foundation, grant No. 19-28323X, for the financial support. R.S., T.P. and D.B. also acknowledge institutional supports RVO: 61388971 and 60077344. Authors thank Dr. Alastair T. Gardiner for his critical reading of the article.

## Author contributions

P.S., T.P., and R.S. designed research; P.S., H.S-M., V.K., D.B., M.M.K., S.L., T.P., and R.S. performed research; P.S., H.S-M., V.K., D.B., M.M.K., T.P., and R.S. analysed the data; and P.S., D.B., T.P., and R.S. wrote the paper.

## Competing interests

The authors declare no competing interests.
