## [Peer Review File · Nature Communications]

Plant LHC-like proteins show robust folding and static non-photochemical quenchingReviewers' Comments:

Reviewer #1:

Remarks to the Author:

LHC-like proteins share common motifs with the antenna proteins of eukaryotic photosynthetic organisms, and many of them are thought to bind chlorophyll and carotenoids, but for most of them, their functions and pigment-binding abilities are not well understood. In this study, they analyzed the pigment-binding ability of two LHC-like proteins from Arabidopsis, LIL3 and ELIP, by using the heterologous express system in the cyanobacterium. The results showed that they can bind chlorophyll and zeaxanthin in the organism. Of course, zeaxanthin is not always present in plants, so it is certain that zeaxanthin is not the genuine pigment they bind, but it is reasonable to assume that the original pigment is lutein.

Purification by sucrose density gradient showed that these proteins are present in dimers. , Ultrafast transient absorption spectroscopy showed that LIL3 and ELIP quench the excitation energy very quickly, which is an important observation since there has been limited information on the quenching of LHC-like proteins.

The manuscript is well-written, and I have no major concern; the discussions are reasonable, and for the most part I agree with these discussions.

Minor points:

1. I'm wondering why they abbreviate xanthophylls or xanthophyll cycle pigments as xant. Could it be a jargon? Why did not they spell out the word?
2. Some labels are missing in Figure 5.

Reviewer #2:

Remarks to the Author:

In this paper LHC-like proteins from plants are investigated. Thus far little is known about the optical properties and exact functions of these proteins. In this work two LHC-like proteins (LIL3 and ELIP2) from plants are expressed in the cyanobacteria *Synechocystis* and purified from there. The pigment composition, absorption/CD spectra and excitation-energy transfer pathways are revealed.

The work is scientifically sound and there is a good back-up with experimental data. I only have problems with the proof provided that ELIP2 is not quenched, e.g. this is hard to see as the sample is heavily contaminated with pigments in DDM (see also below).

Several interesting new findings were reported. I leave it to the editor to decide if they are interesting for a broader audience.

I found the beginning of the introduction a bit confusing as it was focusing on Hlips from cyanobacteria, while the paper is about similar proteins from plants. Maybe it would make it easier if the LHC-like proteins are first introduced and next the cyanobacteria are discussed.

Things that need to be clarified in the text are mentioned below:

"However, as an exception to this rule, there is a family of small (one-helix) Chl 43 binding membrane proteins that shows no similarity to photosystem subunits." In what sense do they show no similarity, functionally/structurally/sequence/...?

"66 Apart from 'true' LHCs, a broad spectrum of so-called LHC-like proteins have been identified," identified in what species? plants and algae?

"In this work, we heterologously expressed and isolated Arabidopsis LIL3.1 (from 79 here on LIL3) and ELIP2 proteins from the cyanobacterium Synechocystis sp. PCC 6803 80 (hereafter Synechocystis)." The proteins are expressed in Synechocystis and isolated from Synechocystis, think the sentence is not correct like it is now, at least one could read that the ELIP2 proteins are from Synechocystis.

I can imagine why the proteins are expressed in cyanobacteria and not in plants, however the reason should be explained in the text.

"Surprisingly, if normalized for the same Chl absorbance and 148 Chl fluorescence emissions of ELIP2 and free Chl in DDM micelles, this protein showed no 149 detectable NPQ (Fig. 1c)." I don't see much of a separation between pigmented ELIP2 and pigments in DDM. For LIL3 the ratio between absorption (DAD) and fluorescence (FLD) is also close to 1, while it is shown to have strong NPQ. So the statement that ELIP2 does not show quenching should be proven or explained better. Maybe a transient absorption or fluorescence lifetime measurements could actually show that ELIP2 is quenched?

2.5 ps in text line 266 and 2.4 ps in table 1.

"Chl phytol chains close the carotenoid-binding 351 cavity (Fig. 5d) and it is very likely that carotenoids also play a critical role in stabilizing the 352 ligated Chl24." Verb is missing in this sentence.

"As in LIL3, the bound (sensed) Chl is efficiently quenched via energy transfer to the S1 state 455 of a carotenoid nearby." To what does (sensed) refer?

Reviewer #3:

Remarks to the Author:

This communication presents a mature study that meaningfully and significantly extend previous works by the authors on the folding and pigment binding of LHC-like proteins as well as on the mechanisms of chlorophyll excitation quenching by carotenoids in these proteins. In the present work, the authors created transgenic cyanobacteria expressing the higher-plant proteins LIL3 and ELIP2, which bear similarity to the cyanobacterial Hlips and are considered ancestral to the plant light-harvesting antenna (LHC) proteins. Using essentially the same methodology as applied earlier with Hlips, the authors purified LIL3 and ELIP2, characterized their oligomeric state, pigment composition and spectroscopic characteristics. Earlier works with in vitro reconstitution have shown chlorophyll binding but exact pigment composition has not been reported. The results convincingly demonstrate that LIL3 dimers stably bind 2 zeaxanthin molecules and 4-5 chlorophylls and that the Chl fluorescence is quenched, presumably by a red-shifted carotenoid. The ELIP2 pigment-protein complex was less stable and showed no red-shifted carotenoid absorption and no fluorescence quenching; however, a chimeric construct Li-ELIP2, containing the N-terminal loop from LIL3 restored the stable pigment binding, red shift and quenching. Ultrafast transient absorption spectroscopy unequivocally confirmed that the chlorophyll excitation quenching in both proteins, LIL3 and Li-ELIP2 occurs via energy transfer to the carotenoid S1 state, as previously found in Hlips. These results could stand on their own, but the authors went further – showing the preferential binding of zeaxanthin over violaxanthin and the ability of the proteins to fold without xanthophylls but not to bind chlorophylls. Finally, they replaced the putative pigment-binding motif in LIL3 that resulted in the expected pigment loss.

These original results represent a considerable experimental effort, are of exceedingly high quality and are obtained with established methodology described with appropriate detail (or references). Much of the experimental methodology is used by the same group(s) in recently published works and is clearly

executed to a high standard. The results are significant in two ways: 1) They confirm a structural motif common to both LHCS and LHC-like proteins that forms a 4-pigment core stabilizing the structure and essential to the assembly of the pigment-protein complex. The authors discuss this in a broader context that provides much insight into the mechanisms of self-assembly of the pigment-protein complexes and even on the evolution of LHCS. 2) The experimental data give strong support to the previously proposed link between the protein-induced twist of the carotenoid ring, the lowering of carotenoid excited-state energy and the activation of energy transfer from nearby chlorophylls, which as a whole provides photoprotective dissipation of energy in the pigment-protein complex. This undoubtedly advances our understanding of the photophysics of carotenoids and nonphotochemical quenching, which are notoriously complex and challenging problems.

As a whole, I find the study very well designed and conducted, the results reliable and the interpretation and conclusions solid and convincing. The paper as whole is well written and logically laid out. There are only a few minor points that could be corrected or improved:

1. The kinetic model of quenching (Fig. 3e) raises some questions. While the kinetic compartments can represent individual pigments in the complex, it is not obvious how the time constants were determined if the compartments are not resolved in the experimental data or in the reconstructed DADS/EADS (Extended data Fig. 7). The experimentally resolved lifetimes do not correspond directly to the eigenvalues of the model. Some more detail on this could be given in extended data.
2. It is not made clear whether only one (half) zeaxanthin is in s-trans configuration and red-shifted (as in HlipC/D) or both. However, the kinetic model assumes fast energy transfer to both – which should necessitate that both are twisted/red-shifted. This could be answered in a future work.
3. It would be useful to show more detailed transient absorption data in extended data – at least time-resolved spectra at more time points or a 2D/3D time-wavelength dependence.
4. Line 161: “de-attachment” - detachment.
5. Line 220: “578 nm” should probably read 587 nm.
6. The panels on Figure 5 lack letters.
7. Line 358: While the UVCD data indicate that LIL3 and LIL3-N177D adopt similar secondary structure, is this sufficient to claim that they are both “fully folded” (implying tertiary structure)? It is hard to imagine that the apoprotein has the same three-dimensional structure as the pigment-protein holocomplex.
8. Line 367: “a1 and a4 Chl” – the numbers don’t correspond to Fig. 5d.
9. Line 370: “the CD spectra strongly resemble the spectrum of monomeric plant LHCII complex” – this is not obvious, certainly not in the Chl Qy region, perhaps some resemblance can be found in the carotenoid region – this should be clarified.
10. Page 16: Isn’t LIL3 necessary for the activity of geranylgeranyl reductase and therefore for chlorophyll biosynthesis? If chlorophyll stabilizes LIL3, hence GGR activity, this makes for a positive feedback regulation – might be worth clarifying.
11. Occasional minor language and typography errors. E.g. compound words like “one-helix”, “high-light-inducible”, “Chl-binding” are missing hyphens; lower-case abbreviations (“xants”) are not standard.

We would like to thank the reviewers for their time spent evaluating and commenting on our manuscript. We thank them for their critical comments that helped us to improve the manuscript. We have considered all comments and suggestions and revised our manuscript as detailed below.

Reviewer #1

The manuscript is well-written, and I have no major concern; the discussions are reasonable, and for the most part I agree with these discussions.

We thank the reviewer for this positive assessment.

1. I'm wondering why they abbreviate xanthophylls or xanthophyll cycle pigments as xant. Could it be a jargon? Why did not they spell out the word?

It is not a jargon. We abbreviated 'xanthophyll' word as it is repeated many times in text. We however agree that this abbreviation can be a bit confusing and, in the revised version, 'xanthophyll' is spelled out.

2. Some labels are missing in Figure 5.

We thank the reviewer for spotting this mistake, it is fixed.

Reviewer #2:

The work is scientifically sound and there is a good back-up with experimental data. I only have problems with the proof provided that ELIP2 is not quenched, e.g. this is hard to see as the sample is heavily contaminated with pigments in DDM (see also below).

"Surprisingly, if normalized for the same Chl absorbance and Chl fluorescence emissions of ELIP2 and free Chl in DDM micelles, this protein showed no detectable NPQ (Fig. 1c)." I don't see much of a separation between pigmented ELIP2 and pigments in DDM. For LIL3 the ratio between absorption (DAD) and fluorescence (FLD) is also close to 1, while it is shown to have strong NPQ. So the statement that ELIP2 does not show quenching should be proven or explained better. Maybe a transient absorption or fluorescence lifetime measurements could actually show that ELIP2 is quenched?

We thank for this comment. We modified text to better explain the principle of NPQ detection during SEC. For all SEC chromatograms, signals of Chl absorbance (DAD) and Chl fluorescence (FLD) were normalized to the same maxima, we apologize for not mentioning it in the original manuscript (it is now explained in Fig. 1 caption, lines 635-636). Although the FLD signal for Chl co-migrating with LIL3 and with DDM is close to 1, the important is actually the ratio between Chl absorbance and Chl fluorescence in LIL3 and DDM peaks. This particular LIL3 prep contained little of free Chl in DDM (less than for Li-ELIP, Fig. 2c), yet the FLD signal is ~3-4x higher than Chl DAD signal in DDM, while for the LIL3 peak this ratio is negative. This is now explained in the text:

Lines 119-121: "The SEC fraction containing LIL3 showed strongly quenched Chl fluorescence when compared with the fluorescence of Chl migrating with DDM micelles, as evident from the ratio of Chl absorbance and Chl fluorescence for LIL3 and DDM peaks (Fig. 1c)."

We admit that the proof of unquenched ELIP2 was rather weak. The ELIP2-pigment complex is apparently unstable during native electrophoresis (Fig. 1b) and the dissociation of pigments probably also occurs during size-exclusion chromatography (SEC; Fig. 1c). We think that a significant fraction of chlorophyll is stripped off the ELIP2 during SEC and runs with detergent. We modified text to make clear that most of free pigments in DDM likely originated from ELIP2 protein and weakened our statement by saying that, using SEC, we found no evidence that ELIP2 is quenched (line 130-132). To address more rigorously the quenching of ELIP2, we have followed reviewer's advice and collected additional transient absorption spectroscopy data of the purified ELIP2. For this particular measurement we omitted sucrose gradient purification step to analyze a very fresh sample; the measurements were carried out almost immediately after the protein elution from the nickel column. The chlorophyll decay kinetics is now added to Fig. 3d. It provides clear evidence that Chl in ELIP2 is not quenched, at least not at the time scale at which the quenching occurs in LIL3 and Li-ELIP. The full dataset, showing that no carotenoid signal is present after Chl excitation of ELIP2, has been added as a new Extended Data Figure 6. We can therefore conclude that the ELIP2 protein isolated in DDM is not in a quenched state.

I found the beginning of the introduction a bit confusing as it was focusing on Hlips from cyanobacteria, while the paper is about similar proteins from plants. Maybe it would make it easier if the LHC-like proteins are first introduced and next the cyanobacteria are discussed.

We intended to describe the whole LHC superfamily from an evolutionary perspective; Hlips are ancestors of eukaryotic LHC-like proteins. Also, Hlips are relatively well studied, including the associated pigments, the mechanism of energy quenching, photoprotective role etc. On the contrary, almost nothing is known about eukaryotic LHC-like proteins. It is indeed possible to introduce first LHC antennas and continue with a short paragraph about structurally related LHC-like proteins. In the rest of introduction, we would however need to describe Hlips anyway, explaining that this or this is relevant to LHC-like proteins and why. We, therefore, believe that to start from Hlips provides evolutionary correct sequence and we thus prefer to keep this order in the Introduction.

"However, as an exception to this rule, there is a family of small (one-helix) Chl-binding membrane proteins that shows no similarity to photosystem subunits." In what sense do they show no similarity, functionally/structurally/sequence/...?

Structurally. The sentence was modified.

"Apart from 'true' LHCs, a broad spectrum of so-called LHC-like proteins have been identified," identified in what species? plants and algae?"

Both in plants and algae. The sentence was modified.

"In this work, we heterologously expressed and isolated Arabidopsis LIL3.1 (from 79 here on LIL3) and ELIP2 proteins from the cyanobacterium Synechocystis sp. PCC 6803 80 (hereafter Synechocystis)." The proteins are expressed in Synechocystis and isolated from Synechocystis, think the sentence is not correct like it is now, at least one could read that the ELIP2 proteins are from Synechocystis.

We agree that the sentence may have caused some confusion. In the revised version the sentence reads:

lines 78-79: In this work, LIL3.1 and ELIP2 proteins from Arabidopsis were heterologously expressed and isolated from the cyanobacterium *Synechocystis* sp. PCC 6803 (hereafter *Synechocystis*).

I can imagine why the proteins are expressed in cyanobacteria and not in plants, however the reason should be explained in the text.

We admit it has not been properly explained in the original manuscript. We have included a short explanation in lines 89-91.

2.5 ps in text line 266 and 2.4 ps in table 1.

Corrected.

“Chl phytol chains close the carotenoid-binding cavity (Fig. 5d) and it is very likely that carotenoids also play a critical role in stabilizing the ligated Chl.” Verb is missing in this sentence.

The sentence was modified: “Chl phytol chains enclose the carotenoid-binding cavity (Fig. 5d) and it is very likely that carotenoids also play a critical role in stabilizing the ligated Chl.”

“As in LIL3, the bound (sensed) Chl is efficiently quenched via energy transfer to the S1 state of a carotenoid nearby.” To what does (sensed) refer?

We discuss that both LIL3 and ferrochelatase could serve as a chlorophyll sensor, but in the revised manuscript we omit ‘(sensed)’ from this sentence for clarity.

Reviewer #3:

As a whole, I find the study very well designed and conducted, the results reliable and the interpretation and conclusions solid and convincing. The paper as whole is well written and logically laid out. There are only a few minor points that could be corrected or improved:

We thank the reviewer for positive assessment of our work.

1. The kinetic model of quenching (Fig. 3e) raises some questions. While the kinetic compartments can represent individual pigments in the complex, it is not obvious how the time constants were determined if the compartments are not resolved in the experimental data or in the reconstructed DADS/EADS (Extended data Fig. 7). The experimentally resolved lifetimes do not correspond directly to the eigenvalues of the model. Some more detail on this could be given in extended data.

As elaborated further in our answer to point 2 below, the focus of the present work was primarily on the biochemistry/molecular biology. The spectroscopy part aimed specifically to demonstrate the existence of Chl quenching by carotenoids. A detailed analysis of the energy transfer pathways is beyond the scope of this manuscript and is still a work in progress. Hence, we include only data using excitation to the main Qy band of Chl, which served well to monitor the overall quenching dynamics. One of our aims was to see whether a unified model for both proteins can be designed. Since analysis in the usual direction, data-DADS-SADS, did not provide unambiguous results, we took a different, rather opposite, direction. We started from the simulated spectra of compartments which are shown below in Figure R1. The carotenoid spectra were obtained from preliminary data measured after carotenoid excitation. These simulated SADS were used to construct datasets corresponding to various energy transfer schemes, which were further analysed in the same way as the experimental data to obtain the DADS/EADS. This showed that it is possible to obtain reasonable fit of the data although some aspects of the dynamics remain unresolved in the simulated dataset. We admit that our model

cannot capture full energy transfer network (which in fact has not been the goal here), but it gives basic picture of quenching dynamics in both systems. The model will serve as a basis to design further experiments, which are currently underway.

Figure R1. Example of simulated data; a) species spectra; b) simulated transient data, incorporate realistic instrument response and “artifacts” around $t = 0$; c) fitted traces. The chosen model is the one from a set of simulations that yielded quantitative agreement with the experimental results while offering adequate description of behaviour of both LIL3 and Li-ELIP complex.

2. It is not made clear whether only one (half) zeaxanthin is in *s-trans* configuration and red-shifted (as in HlipC/D) or both. However, the kinetic model assumes fast energy transfer to both – which should necessitate that both are twisted/red-shifted. This could be answered in a future work.

We thank the reviewer for this comment. Indeed, we have not addressed this issue in the manuscript. However, our preliminary data shown below in Figure R2 demonstrate that, as in Hlips, there are two spectroscopically different carotenoids which could be assigned to *s-trans*/*s-cis* configurations of one of their terminal rings. Comparing these data (maxima of the carotenoid S1-S_n bands) with those shown in Fig. 3 in the manuscript, it is obvious that the red-shifted carotenoid is the (main) quencher as it has been reported for Hlips in Ref. 5.

Figure R2. Transient absorption spectra at 2 ps after direct excitation of carotenoids in LIL3 (left) or Li-ELIP (right). Excitation wavelengths are 540 and 500 nm for LIL3, and 530 and 490 nm for Li-ELIP.

The reason why we have not included these data in this manuscript is that the situation is more complicated and requires further experiments and analyses. It is obvious from our preliminary data that the spectral distinction of carotenoid is very clear in LIL3 (essentially mirroring the situation in Hlips reported earlier), but it is much less pronounced in Li-ELIP. Carotenoid-to-chlorophyll energy transfer is also different in these two complexes, as evident from amplitude of the Chl bleaching band at 2 ps after carotenoid excitation, further complicating the situation. Analyses and discussion of this

problem would enormously expand the manuscript. Thus, we think this issue deserves a separate study (as in fact suggested also by the reviewer). We also plan to include at least two different excitations of the chlorophyll band, because Li-ELIP has broader Chl-a band indicating also spectrally different chlorophylls. Along the same lines, we have also measured (on request of Reviewer #2) transient absorption data after Chl excitation of unquenched ELIP2 (see panels c and f in new Extended Data Figure 6). Information about carotenoid-chlorophyll energy transfer in unquenched ELIP2 will further expand the matrix of data (we do not have these data yet), which could be used to obtain a more comprehensive picture of energy transfer network between carotenoids and chlorophylls in LIL3 and Li-ELIP. This is still work in progress. We believe this complete analysis based on multiple transient absorption datasets deserves a separate study and is beyond the scope of this manuscript. Here, we unequivocally demonstrate the energy transfer quenching of chlorophylls in LIL3 and Li-ELIP and we propose a model assuming a single (mean) chlorophyll-carotenoid quenching rate, though it is possible there are more quenching pathways. Identification of individual inter-pigment energy transfer rates, as well as (spectral) identification of chlorophylls that are preferentially quenched, will be a subject of a future study.

3. It would be useful to show more detailed transient absorption data in extended data – at least time-resolved spectra at more time points or a 2D/3D time-wavelength dependence.

We have added the requested data to Extended Data section. New Extended Data Figure 6 shows both full datasets as 3D plot and selected transient absorption spectra at a few delay times. On request of the Reviewer #2, we have also added new data measured for unquenched ELIP2. This new dataset is also shown in the new Extended Data Figure 6.

4. Line 161: “de-attachment” - detachment.

Corrected.

5. Line 220: “578 nm” should probably read 587 nm.

Corrected.

6. The panels on Figure 5 lack letters.

Corrected.

7. Line 358: While the UVCD data indicate that LIL3 and LIL3-N177D adopt similar secondary structure, is this sufficient to claim that they are both “fully folded” (implying tertiary structure)? It is hard to imagine that the apoprotein has the same three-dimensional structure as the pigment-protein holocomplex.

Yes, the reviewer is correct. The structure of apoprotein can be hardly identical with a pigment-protein complex. In contrast to plant LHCs, the pigment-less LIL3-N177D is apparently very stable, which is remarkable. Plant LHC proteins were studied extensively using CD and it is well known that their secondary structure (α -helix) formation is tightly coupled with pigment binding. We used UV CD to show that the secondary structure is established in LIL3 even without pigments, the revised version is accurate in this point; lines 282-284.

8. Line 367: “*a1 and a4 Chl*” – the numbers don’t correspond to Fig. 5d.

We modified the sentence to explain better that the Fig. 5d shows one lutein binding site only (*lut1*); Chl *a4*, located symmetrically to *a1*, is not depicted. The new version reads:

Lines 291-294: In addition, in all type LHCS with solved structure, two xanthophyll molecules can be always recognized in cross-braced *lut1* and *lut2* binding sites with their isoprenoid chains wrapped up by phytol tails of *a1* (*lut1*) and *a4* Chl (*lut2*; see Fig. 5d for the *lut1* binding site).

9. Line 370: “the CD spectra strongly resemble the spectrum of monomeric plant LHClI complex” – this is not obvious, certainly not in the Chl Qy region, perhaps some resemblance can be found in the carotenoid region – this should be clarified.

Admittedly, the formulation was not very clear; the statement indeed referred specifically to the spectral region of Chl *a* Soret band and carotenoids. This was now made clear in the text and the assertion was explained in more detail; lines 297-298.

10. Page 16: *Isn’t LIL3 necessary for the activity of geranylgeranyl reductase and therefore for chlorophyll biosynthesis? If chlorophyll stabilizes LIL3, hence GGR activity, this makes for a positive feedback regulation – might be worth clarifying.*

We thank the reviewer for this comment. Yes, we indeed propose a positive feedback regulation. It is known that LIL3 is abundant in etioplasts although they are devoid of chlorophyll (Reisinger et al. FEBS Lett. 582:1547, 2008). During the early stage of photomorphogenesis, first chlorophyll produced in etioplasts seems to bind to LIL3 (Reisinger et al. FEBS Lett. 582:1547, 2008), which could activate GGR for the synthesis of a large quantity of phytol required for the building of thylakoid membranes. We have re-written this part of the discussion to clarify this point; lines 365-370.

11. *Occasional minor language and typography errors. E.g. compound words like “one-helix”, “high-light-inducible”, “Chl-binding” are missing hyphens; lower-case abbreviations (“xants”) are not standard.*

Corrected. We still use *xant-* for *Synechocystis* mutants lacking xanthophylls following a standard nomenclature for bacterial mutants

Reviewers' Comments:

Reviewer #1:

Remarks to the Author:

This study is scientifically sound, and the manuscript is well written.
I do not have further comments on the revised manuscript.

Reviewer #2:

Remarks to the Author:

All my concerns are now addressed so the manuscript can be accepted.

Reviewer #3:

Remarks to the Author:

I am pleased to see that the authors have carefully answered all questions and comments, added missing experimental data in Extended Data and corrected other lapses in the revised version. Originally these were of minor importance anyway, I reiterate that the bulk of the experimental results are novel, of high quality and standard and clearly support the conclusions. This is a significant contribution that should be published.